# From Monitoring to Assisting: A Systematic Review towards Healthier Workplaces

**DOI:** 10.3390/ijerph192316197

**Published:** 2022-12-03

**Authors:** Laís Lopes, Ana Rodrigues, Diogo Cabral, Pedro Campos

**Affiliations:** 1Interactive Technologies Institute, Laboratory of Robotics and Engineering Systems (ITI/LARSyS), 9020-105 Funchal, Portugal; 2Faculty of Exact Sciences and Engineering, University of Madeira, 9020-105 Funchal, Portugal; 3Instituto Superior Técnico, University of Lisbon, 1049-001 Lisbon, Portugal

**Keywords:** stress, workplace, affective computing, embodied interaction, tangible interfaces

## Abstract

Long-term stress is associated with a decline in global health, affecting social, intellectual, and economic development alike. Although comprehensive action plans have been implemented to provide people access to mental health services and promote mental well-being, employees’ mental health generally takes second place to productivity and profit in business settings. This review paper offers an overview of the current interactive approaches used for relieving work-related stress associated with mental health. Results from the 38 included studies show that affective computing is used mainly for monitoring purposes and is usually combined with tangible interfaces that collect workers’ physiological changes. Although the ability to sense and predict employees’ affective states can potentially improve mental health in the workplace, there is a substantial disparity between monitoring one’s health and the delivery of practical interventions to mitigate stress found in the surveyed studies. Designing systems that capitalize on embodied interaction principles is paramount, especially in the post-pandemic context, as the concepts of physical and mental safety take on new meanings that must be consciously and carefully addressed, particularly in workplace settings. Finally, this paper highlights the main design implications for the effective implementation of interfaces to help mitigate stress in the workplace.

## 1. Introduction

Chronic stress is a crucial contributor to global health decline. In addition to its negative effects on physical health, chronic stress also increases the likelihood of developing more serious psychological disorders, such as anxiety and depression.

In workplace environments, stress can emerge from a variety of factors and translates into cognitive impairments (e.g., lack of focus and irritability), which affects employees’ health, morale, and productivity. In addition, it leads to an increased risk of cardiovascular disease, especially when employees feel a lack of control over their work, and are not being sufficiently compensated for their efforts, or are unable to take part in the decision-making process [1,2,3,4].

Although job productivity has increased with remote work during the COVID-19 pandemic, so too has perceived stress [5]. Although several actions have been proposed to alleviate work-related stress and increase mental well-being [6,7], mental health remains largely overlooked in business settings, despite its clear influence on social, intellectual, and economic development [3].

However, with the growth of mobile health (mHealth), new technologies have emerged that allow people to monitor, record and assess their health information, and several mobile applications have been designed to help users cope with stress.

Although a vast body of literature exists on the importance and potential of using multimodal sensing technologies to reduce stress, and on the different types of systems and devices and their trade-offs regarding sensing affective-related data [8,9,10,11,12], to the best of the authors’ knowledge, no review has been conducted on the different technological approaches used to design interactive systems and interfaces specifically to help mitigate stress in the workplace. This work aims to fill this gap.

The aim of this review paper is to assess how affective computing and tangible interfaces have been used thus far to monitor and deliver interventions that help manage workers’ mental well-being in business settings and highlight possible future development opportunities in the area.

This paper is structured as follows. Section 2 reviews some of the existing literature on mental-health monitoring systems and devices, and highlights their key findings in relation to mitigating stress in the workplace. Section 2 also offers an overview of work-related stress and introduces the key concepts of affective computing, tangible interfaces and embodied interaction, and considers their significance in the design of interactive systems and interfaces for fostering healthier work environments. Section 3 describes our research methodology, and Section 4 presents the results of our review. Finally, in Section 5 this work discusses how the existing affective and tangible interfaces can inform the design of new technologies to help mitigate stress and foster healthier work spaces.

## 2. Background and Related Work

As previously stated, information systems and technologies have enabled the assessment and delivery of on-site interventions for stress management. However, as shown by previous studies, the majority of systems and interfaces are designed primarily for monitoring purposes.

Gimpel et al. [11], for example, analyzed the current trends of mobile stress assessment (MSA), proposed a taxonomy for the existing systems, and found that the majority of studies fell into the Visible Tracker category, which employ intrusive (i.e., visible to the user) sensors to collect biological data in order to assess whether or not people are stressed. Additionally, because such MSA systems also require access to aggregated personal data, they raise privacy concerns that must be taken into account (as it is still possible to identify individuals via thorough data analysis), especially when designing for mental well-being applications.

Similarly, Garcia–Ceja et al. [9] reviewed the mental health monitoring systems (MHMS) literature and proposed a classification system composed of three categories: study type, duration, and sensing types. Although works aimed at other mental health conditions (e.g., anxiety and depression) look to predict, or at least examine the relationship between variables, studies related to stress focus only on mental health recognition.

The development of wearable sensors has rendered the detection of users’ affective states in situ a simpler task. Nevertheless, doing so requires access to data that can be considered highly invasive. The results reported by Garcia–Ceja et al. [9] show that although some sensors used to measure stress levels are unintrusive (e.g., accelerometers), they are often combined with others that raise substantial privacy issues (i.e., call logs, audio, video, and GPS).

Notwithstanding the validity of participant consent and ethical approvals required for collecting affective-related data, the deployment of such systems on-site can further induce stress in workers, who view their privacy as threatened when wellness programs are imposed by companies who require them to use wearables in the workplace [10,13].

Khakurel et al. [10] analyzed how wearable devices have been used in the workplace, identified several categories of wearables (e.g., smartwatches, heart rate monitors, etc.) and expanded on the advantages and challenges those can bring to the work environment. Similarly, in [12], Sharma and Gedeon provide a review of the most common unobtrusive sensors (i.e., that can be used regularly without interfering with daily activities) and techniques used to detect and model stress.

Although wearables allow employers to monitor work-related stress, assess the moods of their employees, assist in physically demanding jobs (thus improving worker safety), help track sedentary behaviors, and deliver just-in-time content and interventions, some of the challenges reported in [10] are related to the technology itself (e.g., accuracy and battery issues), and the social and ethical concerns that arise from having access to large amounts of health-related data, such as employees’ concerns about whether or not their privacy will be violated and how the collected data will be used.

### 2.1. Work-Related Stress

In today’s highly competitive society, job insecurity has become a real concern with workers now more than ever pressured into being increasingly productive. If, prior to the pandemic, spending more time at the office took a toll on employees’ physical health (e.g, by prolonged sedentarism or excessive heavy-duty strain), remote work has increased perceived stress even more, especially in men [5].

Although a number of researchers have investigated the benefits (e.g., savings and increased productivity) and disadvantages (e.g., isolation and lack of support) of technology-aided remote work for both companies and employees [8,14], Mann’s work showed that productivity increases when telework is performed by choice, and although teleworkers usually feel less stressed than office workers, they do tend to experience more negative emotions [15].

Having little to no work–family boundaries, a lack of control over one’s work and feeling excluded from the decision-making process, especially in the current climate, contributes to an increase in occupational stress, which can affect one’s ability to accurately assimilate cognitive, affective, and autonomic brain responses in the long-term, and can be manifested in symptoms such as memory loss and lack of focus, leading to an increase in health risks and mental health impairment [1,2,3].

Although the aim of this paper is not to assess the conditions and effects of the COVID-19 pandemic, one must be aware that businesses and employees are facing damaging effects, both economic and psychological, and that the concepts of physical and mental safety are taking on new meanings [16], which must be consciously and carefully addressed.

Despite the fact that various actions have been proposed to alleviate work-related stress and increase mental well-being [6,7], mental health is still largely overlooked in business settings. With this in mind, the authors set out to review the last 10 years of existing literature focused on the development of embodied, affective, and tangible interfaces to monitor, evaluate and positively influence workers’ physical health and affective states.

### 2.2. Affective Computing and the Workplace

The ability to sense and assess people’s psychological states not only makes instant feedback and in situ interventions possible [17], but has the potential to sense employees’ emotions towards workload, monitor their physical activity and deliver interventions to increase well-being, which in turn can help improve workers’ mental health states. Nevertheless, as Picard [17] stated, affective monitoring in the workplace can also be perceived as intrusive, especially when used to detect employees’ productivity and response to workload.

However, it is possible to develop and deploy seemingly fair monitoring systems. A study conducted by Moorman and Wells [18], for example, highlights the importance of designing systems that offer constructive feedback and allow employees to decide how the collected information is interpreted. According to the World Health Organization [2], employees who feel in control over their work load and environment are not only healthier but also more productive, as positive affect has been shown to enhance cognitive processing (i.e., attention, decision making) and encourage conflict resolution, creativity and healthier interpersonal relationships [19].

Therefore, when designing new emotion-sensing technologies, it is crucial to take into account how the pervasiveness of such devices will be perceived by employees and, more importantly, how such interfaces can help create a more constructive work environment and facilitate employee empowerment.

### 2.3. Embodied Interaction and Tangible Interfaces

The recognition and capitalization of social aspects and physical skills are crucial to the design of embodied interactions as the social context shapes how people engage in real-world exchanges and gives meaning to the way people interact through technology [20]. This phenomenological approach helps researchers better understand people’s activities and surroundings, which allows them not only to plan how to better introduce new technologies into people’s daily lives but also informs the design of more meaningful and productive experiences and artifacts.

Not only is this notion of embodiment intertwined with social computing, and the manner in which current businesses operate, it is also a cornerstone of tangible interfaces, since only by acknowledging the importance of intrinsic physical affordances in the world is it possible to exploit them and design truly ubiquitous interactions between people and (otherwise limited) data.

Fidgeting, for example, represents a common form of emotional self-regulation that can be capitalized as an unobtrusive means of collecting affective information and delivering stress-mitigating interventions through the use of easy-to-reach objects that can be found on employees’ workstations [21,22]. The work of Karlesky and Isbister [21] not only highlights the importance of applying embodied interaction principles to the design of new emotion regulation interfaces, their results are also in line with this article’s findings, namely that tangible interfaces are a promising yet underexplored area of research for the delivery of stress-mitigating interventions, especially in the workplace.

## 3. Methodology

The aim of this paper is to investigate how tangible interfaces, affective computing and embodied interaction principles have been employed over the last decade to help mitigate employee stress in the workplace.

To do so, a systematic review was carried out according to the Preferred Reporting Items for Systematic Reviews and Meta-Analyses (PRISMA) guidelines [23], where the following scientific repositories and databases were searched for research articles and journal papers published during the last 10 years: ACM Digital Library (The ACM Guide to Computing Literature) and IEEE Xplore. Both portals cover most of Computer Science and Human–Computer Interaction (HCI) journals and conference proceedings. Although reviewing psychology journals would produce interesting research, this work aims at identifying technological trends in monitoring and mitigating stress in the workplace, and thus focuses on Computer Science and HCI literature.

The search was conducted by combining the following search terms: Keyword:(stress OR anxiety OR mental health OR wellbeing OR well-being) AND Keyword:(work* NOT workshop NOT workload OR office OR business OR job OR occupational OR corporat*) AND Abstract:(intervention OR manage* OR regulat* OR sensing OR measure* OR recognition OR detection) AND Abstract:(tangible OR affective OR embodied OR interact*). To ensure the accuracy of the search process, different spellings were intentionally used, as well as the asterisk (*) in the terms indicating a wildcard.

Additionally, to achieve a more manageable sample size, the first two search expressions are related to terms found within Authors’ Keywords, and the remaining terms are searched under the Abstract data field. For example, ACM Digital Library retrieved 162,567 results prior to this limitation, and 33 after.

The output of records were then examined taking into consideration the title of the article as well as the abstract. From 139 screened records, 34 were assessed for eligibility and 18 studies were included in this review. The initial search was completed in June 2022 and the search process and subsequent examination lasted from June to August 2022.

### 3.1. Inclusion and Exclusion Criteria

To be included in this review, records needed to be related to the development or use of embodied, affective or tangible interfaces for predicting or delivering interventions that focus on mitigating stress-related problems in work environments. Records needed to use at least one of the aforementioned approaches to be included. The publication date should be between January 2012 and June 2022.

Records that were related to job burnout and worker performance from a productivity-only basis were excluded, as well as articles in which the work environment was overly specific (e.g., [24]). Literature reviews, meta-analyses, book chapters, Extended Abstracts, Work-in-Progress and other papers that were not directly related to the development or evaluation of new interfaces or devices to mitigate work-related stress were also excluded from this review.

### 3.2. Studies Found

From 162,567 records found in the previously mentioned databases, 139 peer-reviewed research articles were screened and 34 assessed for eligibility by two of the authors (A.R. and L.L.) by reading through the studies. From those, 18 were found to meet the inclusion criteria. Inter-rater reliability was determined using the Cohen’s kappa coefficient (k≈0.82).

A backward search [25] was conducted, where the most relevant citations from the included studies were scanned through, and 46 records were found. From those, 33 did not meet the inclusion criteria and 5 duplicate were excluded. A forward search [25] was also performed, using Web of Science, to ensure all relevant studies were included into the final eligibility round of this review, and 12 more studies (out of 31) were added to the review. At the end, a total of 38 studies were included in this review. Figure 1 illustrates the detailed PRISMA [23] flow diagram used to describe the final search process.

## 4. Results

Two of the authors (A.R. and L.L.) independently coded each of the 38 included studies into 4 main concepts—Tangible, Affective, Mobile and Embodied. Studies that used Tangible, Affective, and Mobile interfaces were divided into monitoring and intervention delivery, as shown in Table 1, and the ones who used Embodied interaction principles were coded only when the authors of the study provided clear evidence of having used phenomenological frameworks during their research or other relevant ethnographic methods to inform the design of their systems. The Cohen’s kappa coefficient was used to assess inter-rater reliability (k≈0.95). Despite the fact that the term tangible is typically used to describe interactions with artifacts that can be physically manipulated in space, the examined literature retrieved results focused mainly on wearable interfaces.

### 4.1. Monitoring

The retrieved results show the most used approach to monitor workers’ stress levels is affective computing, which is often (73.5% of the time) combined with tangible sensing, such as wrist- and chest-worn devices, to mainly monitor workers’ electrodermal and physical activity.

According to the surveyed studies, combining affective monitoring, through the use of surveys and questionnaires, and tangible interfaces allows for the unobtrusive collection of behavioral and physiological data (such as physical and electrodermal activity) and arousal estimates. These measurements are used to not only collect ground truth information from workers, but also to create conventional Machine Learning (ML) models to detect stress levels in business environments (e.g., [29,32,41,63]).

Since stress prediction can be identified as a classification problem, the most common modelling techniques employed by the retrieved studies are Support Vector Machines (SVM) [27,29,41,54], Artificial Neural Networks (ANNs) [27,32,39], Decision trees [35,36,48], and Naive Bayes classifiers [29,35]. These results are in line with those reported by Sharma and Gedeon in [12].

The sensing modalities found within the studies, as proposed by Garcia–Ceja et al. in [9], include software sensing (e.g., computer logging [41,44,61]), external sensors (such as cameras [32,41,43,58] and mouses [33]), and wearable sensors, which can be divided into smart watches/bands, electronic textiles (such as the OMsignal smart shirt used in [60], and flexible (i.e., attached to the skin for monitoring purposes, such as the the BioGraph Infiniti Biofeedback sensors used in [56]).

Although smartphones were considered a type of wearable sensing device by [9], we chose to make a visible distinction between the two in our results (as shown in Table 1), since there is already an extensive body of literature on mobile health (mHealth) systems.

Although a high number of studies were expected to use tangible artifacts (i.e., those that can be physically manipulated in space) to monitor stress in work environments, since they are readily available and unobtrusive, the most common sensing type found within the included studies was wearables, especially off-the-shelf wrist-worn devices [43,44,49,50,52,53,57,58,60,61,62]. These results are in line with those of Garcia–Ceja et al.’s review [9], where they found stress-detection studies always employ some form of wearable sensors, as they are unobtrusive and enable researchers to easily track and identify individual subjects while offering the same sensing possibilities as external devices.

In addition to sensing behavioral and physiological data, most studies (27 out of the 34 used to monitor worker stress) also required users to complete surveys. Although self-reports are a widely accepted measure of stress, they require a great deal of attention from the user and rely on highly subjective accounts that can be affected by how people recall past experiences [50].

Some of the most common standard scales (i.e., employed by four or more studies) used to measure stress and mental well-being are the Positive and Negative Affect Schedule (PANAS) [45,53,59,60,63], the Perceived Stress Scale (PSS) [38,40,52,54], and the Visual Analogue Scale (VAS) [38,41,47,48].

The PANAS [64], a well-established method for evaluating self-reported stress, was developed to assess individuals’ mood in relation to positive (PA) and negative affect (NA), where individuals who display high PA are more easily engaged and focused, while high levels of NA is generally a good indicator of fear states and stress. Although the Perceived Stress Scale (PSS) was most commonly used to measure baseline stress levels during the initial phase of the projects, the Visual Analogue Scale (VAS) was used pre- and post-test, to measure the participants’ changes in perceived stress levels.

Even though questionnaires and daily self-assessment tools were used to help researchers better understand and label acquired data [26,27,28,30,37,43,44,46,49,50,56,57], only four of the included studies showed clear evidence of using embodied interaction principles to help guide their research [26,27,37,63], presented below.

#### 4.1.1. A Phenomenological Approach and User Requirements

Observing the existing literature on occupational well-being and combining it with user research in order to understand employees’ work styles and needs allowed Koldijk [26] to formulate a set of design requirements for the development of a system to help mitigate knowledge workers’ stress. Similarly, in their work, Park et al. [27] detailed some user requirements for the design of systems to help improve knowledge workers’ mental health: (a) physiological monitoring devices should not hinder users’ work, (b) systems should provide workers with access to their physiological data, (c) users should be able to voluntarily request stress-mitigating interventions, and (d) collected data should be protected.

Such requirements are in line with the work of Xue et al. [65], who noted that the development of mental health monitoring systems raises important ethical questions and privacy concerns (e.g., related to employee anonymity and how employers may handle or possibly mismanage their personal data). Moreover, the aforementioned requirements also align with those reported by Moorman and Wells [18], which highlights that interfaces should be developed in a way that enables employees to not only receive constructive feedback but also dispute it in real time.

By employing an experience sampling methodology (ESM) in a software company, Kuutila et al. [37] were able to gather insights into the relationships between employees’ affective states and the context of their work. This was achieved by linking subjective and self-reported affective data, to objective variables (such as time pressure and daily number of chat messages). Since the social fabric surrounding an individual presents a major contributor to mental well-being, only by empowering workers and providing them with a sense of inclusion is it possible to create healthier business environments [2]. Moreover, employing a phenomenological and qualitative research approach prior to the development of new systems is crucial to the understanding of people’s lived experiences within an occupational context.

#### 4.1.2. Sensor Trends

As exhibited in Figure 2, physiological measurements of stress-related data were used more frequently than non-physical ones to monitor workers’ mental states, as they allow researchers to detect emotional arousal and collect ground truth information that can be compared against self-reported data retrieved from surveys and questionnaires.

The prevailing physiological signals employed to measure stress in the workplace exploit the electrical conductance of skin (SC, EDA and GSR) [29,32,38,41,42,49,52,55,56,58,61,62], Heart Rate Variability (HRV) [32,33,38,39,40,41,42,44,50,51,54,59,62], and Heart Rate (HR) [38,40,43,51,54,55,56,58,60,61]. These are in line with Sharma and Gedeon’s findings [12], as electrodermal activity and HRV are unobtrusive indicators of autonomic nervous system activity and can be used to predict mental disorders and relaxation states [59]. However, in real-world settings, Heart Rate Variability can provide more accurate results than EDA, as the latter can be influenced, for example, by external environmental changes [50].

Compared with other physiological stress measures that can be obtained from wearable sensors in daily life, HRV is more reliable in real-world settings (outside the laboratory). For example, skin conductance (i.e., electrodermal activity [EDA]) can be difficult to measure in dry, indoor air-conditioned settings as the electrodes rely on sweat to measure conductance. In addition, some people naturally do not produce adequate EDA signals [33].

The most common off-the-shelf devices used in the included studies to monitor users’ electrodermal activity and Heart Rate Variability were external sensors developed by TMSI—since some of the retrieved results [29,32] used data from the SWELL-KW dataset developed by Koldijk et al. [26,41]—, Empatica E4 [47,49,52,58], and chest-worn heart rate monitors from Zephyr [44,54,58].

Non-physiological measurements, on the other hand, rely mostly on accelerometer data [35,36,46,52,58,59,60,61,63] and computer logging tools. Although Mark et al. [45] state that physical activity data are not a good indicator for stress prediction, since measurements related to heart rate variability and respiratory rate usually exceed those collected by smartphone actigraphs or other (more intrusive) devices [59], the work of Vadym [66] shows that combining both types of HRV signals with accelerometer data can improve stress recognition since physical activity can increase heart-rate levels and galvanic skin response, and thus hamper results when only one measurement modality is used.

Although computer logging is a non-invasive measurement of stress, especially in the work context, only one of the included studies modified peripheral devices to unobtrusively monitor participants’ affective states [33], which demonstrates a clear gap in the use of tangible interfaces for stress detection in the workplace.

### 4.2. Intervention

#### 4.2.1. Virtual Reality-Based

From the 38 studies that met the inclusion criteria, less than were 20% found to deliver stress-mitigating interventions, of which only 3 used tangible interfaces to help workers cope with stress, and with all of them employing head-mounted displays to deliver Virtual Reality (VR) interventions [34,55,56].

In the work of Thoonde and Oikonomou [34], a Virtual Reality application was developed to foster calming states. Their application uses Oculus Rift to display the virtual environment, which was designed so that different objects in the scenario could provide users with work-related information. Their findings show a positive user response towards implementing this approach in the workplace, and suggest VR can also help increase workers’ focus. However, their experiment was conducted in a controlled environment, and thus it is not possible to ascertain whether such an approach would be feasible in a business environment.

While the work of Thoonde and Oikonomou [34] focused on the design and evaluation of an intervention alone, Ladakis et al. [55] employed a wrist-worn device and a Moodmetric smart ring to monitor users’ HR and EDA in order to assess the efficacy of their VR approach. They developed a serious game to help mitigate stress via deep breathing exercises which were delivered to the user using the Oculus Go, a more affordable alternative compared to the device used by [34].

Although work has been conducted on the use of virtual reality systems to help treat and relieve stress, and with both of the previous approaches receiving a positive response towards implementing VR-based interventions in the workplace, their findings rely on qualitative, self-reported results. Moreover, VR-based approaches may not represent the best option for delivering interventions in business environments, since, as the work of Abdul Manaf et al. [56] demonstrates, there are no significant differences between immersive and non-immersive (i.e., smartphone-based) stress-reducing techniques, as long as the presented content is nature-related and fosters calming states.

#### 4.2.2. Smartphone Apps

Additional mobile interventions were also evaluated by two other included studies [57,63]. Bostock et al.’s work [57] showed that participants who completed at least 10 meditation sessions using the mindfulness meditation app, Headspace, presented significant improvements in daily positive affect and global well-being, and that meditation can also have a positive impact on workers’ perception of job control, which, as previously stated, is one of the key aspects that can contribute to occupational stress.

While the work of [57] focused on the evaluation of an existing app, Paredes et al. [63] developed a mobile recommender system that uses Machine Learning (ML) to deliver personalized micro-interventions to workers. The mobile app collected data for the ML algorithm which, based on individuals’ personalities and context, prompted users to complete one of four intervention types: positive psychology, cognitive behavioral, meta-cognitive, or somatic, and each of the categories featured an individual and a social intervention. Their results reveal that the somatic interventions that can be completed on one’s own (e.g., stretching) are not only preferred but also more effective at mitigating stress, followed by individual positive psychology, whereby users are asked to reflect upon their strengths, and by social somatic interventions.

#### 4.2.3. Desktop Applications

The remaining two studies that used interventions to help reduce stress in the workplace were conducted by Sano et al., where desktop applications to mitigate occupational stress were developed [53,54]. In [53], Sano et al. collected affective and physiological data from technology company workers, and developed a desktop application to provide employees with stress mitigating interventions and advice. Their findings are in line with the results from [63], and show somatic-based approaches (e.g., prompting workers to take a short walk with a colleague or improving their posture) are the preferred and most effective type of intervention. However, in [53] Sano et al. note that the timing at which the interventions were delivered was considered inopportune by users more than 50% of the time. In order to resolve this issue, and to help the design of future just-in-time interventions, Sano et al. [54] conducted an investigation to identify the optimal time to deliver micro-interventions during business hours. Their findings show participants preferred, and voluntarily requested, interventions during early morning and early afternoon (i.e., before becoming immersed in their work), and considered the interventions as better when they were voluntarily solicited. Somatic interventions were the most effective, and social somatic interventions (e.g., walking with someone) showed higher self-reported stress reduction. Contrary to [63], the least effective and least preferred type of intervention was positive psychology (e.g., making someone else feel better).

The results from such interventions (summarized in Table 2) provide valuable insights to the design and development of new interfaces aimed at mitigating occupational stress in the workplace, discussed later on in the Design Implications subsection.

## 5. Discussion

This study maps the work-related stress research produced over the last 10 years, and contributes to the HCI community by (1) presenting the trends that arose from the selected studies and (2) outlining gaps in the literature and highlighting design implications for the development of novel stress-mitigating systems in the workplace.

Although extensive research has been conducted to identify workers’ emotional states, our results show there is still a significant disparity between monitoring employees’ stress levels and delivering practical interventions to help mitigate stress in the workplace. Additionally, the fact that less than 11% of studies reported using embodied interaction principles to guide the design of their systems presents a serious concern, since context is a critical part of informing the design of new interfaces aimed at improving mental health, especially in business environments.

Similarly, as demonstrated in Figure 3, 15 of the 37 included studies that used or collected data from participants to either train Machine Learning (ML) models or deliver stress-mitigating interventions to workers failed to provide a detailed description of the sample from which data were collected. From the retrieved studies that accurately described their sample, 45% used data collected from European countries [26,29,32,35,36,37,41,43,51,57], of which half either reported the development of or employed the SWELL Knowledge Work dataset (SWELL-KW) to train ML models [26,29,32,41,43]. However, according to a 2022 global survey from GALLUP [7], employees from Europe tend to experience less negative emotions (such as worry and stress) than those from the United States or East Asian countries, for example. This infers that ML models trained with datasets from other locations or results reported from a single geographical location may not be replicable or representative of the global population.

Another important issue that arose in this review were privacy-related concerns towards how employers may handle and possibly misuse employees’ personal data. As businesses continue to grow into a productivity-driven model, it is paramount to guarantee and safeguard workers’ privacy, especially with specific emphasis being placed on mental health issues.

According to the World Health Organization [2], not only do material security and environment conditions influence employees’ emotional states, but their social support structure is also a major contributor to their mental well-being. However, as the works of Lupton [13] and Khakurel [10] demonstrate, the implementation of wellness programs in the workplace still prioritises business interests over the mental health of employees, who are often required to surrender access to their private personal data that, even if anonymized, raises serious privacy issues regarding how data is stored within the company, who is granted access to it, and how will the sensitive data be managed. Consequently, the clear lack of transparency fosters employee distrust and poor work environments.

Finally, because in the mental health domain there are no one-size-fits-all solutions, employing a phenomenological approach to the design of such interfaces is necessary to help researchers understand not only workers’ concerns regarding how their collected data will be used and treated, but also the user requirements, which are paramount in a post-pandemic context where the concepts of both physical and mental safety are adopting new and important meanings that warrant careful addressing in order to create healthier business environments.

### 5.1. Limitations

Although this paper aims to review the current technological (i.e., embodied, affective, and tangible) interfaces used to monitor and manage work-related stress, not having extended our initial search to include psychology journals may have limited the number of studies included in this research, since the social sciences have relevant venues dealing with the assessment of stress-related interventions. However, some of those records were recovered during the backward and forward stages of the search process. The retrieved studies were published in conferences such as CHI and Affective Computing and Intelligent Interaction (ACII), and journals such as MDPI Sensors and Journal of Occupational Health Psychology. Thus, we believe that the retrieved results are representative of the HCI literature in dealing with the monitoring and development of interfaces aimed at relieving stress in the workplace.

Additionally, owing to the vast comprehensive nature of the embodied interaction discipline in addition to the review and coding of the studies being based on the authors’ understanding of the term, bias might have been introduced in this review. Although we acknowledge the limitations of our review, we were still able to find trends in the use of tangible, which as the results demonstrated, are mostly wearable interfaces and affective computing systems developed to help assess and mitigate work-related stress. In addition, we were able to translate these findings into design possibilities for HCI researchers developing mental health technologies in the occupational space.

### 5.2. Design Implications and Future Directions for Research

The main design implications that arose throughout this study concerning the development of new interfaces aimed at mitigating work-related stress are presented next.

Working with standard methodological approaches is pivotal to ensure the reported results can be accurately compared across different studies. Researchers’ lack of transparency when reporting results can delay and potentially hamper the development of the field, since findings may not be significant or even reliable to inform and guide the design of new interfaces aimed at relieving work-related stress. This is important not only when reporting methods and procedures—to ensure the research is replicable—but also when describing samples, since employees of different nationalities may perceive and deal with negative emotions in a particular way.

The trade-off between collecting personal data and delivering mental health interventions remains unbalanced, as most of the retrieved studies focus entirely on monitoring stress-related signals. A possible solution to make use of the collected data is to apply machine learning models to predict the optimal time to deliver personalized stress interventions

Mental health interfaces should enable employees to receive constructive feedback and, perhaps more importantly, dispute it in real-time. Such features help increase user engagement and empower workers by supporting them in reaching their goals.

Timely somatic interventions and embodied interactions that require workers to shift their focus to their bodies are effective and recommended. Mindfulness practices and stretching exercises, for example, can help release physical tension and disrupt ongoing stress patterns. However, when designing such interventions, researchers should make sure they do not interrupt employees’ workflow.

Tangible artifacts and peripheral devices are still largely overlooked when it comes to the development of novel technologies to monitor and assist workers’ mental well-being. Although commercial wearable sensors are considered tangible interfaces, only a few included studies reported the development of custom-made artifacts to monitor stress signals [33,46]. This may be due to the social acceptability of such devices. However, in business settings, peripherals and other office equipments can provide ubiquitous access to data that can be used to infer one’s mental states (e.g., [67,68]) and potentially deliver stress-mitigating interventions.

## 6. Conclusions

A systematic review of the literature was performed to assess how tangible interfaces and affective computing have been used to help mitigate stress in the workplace. The retrieved results show that wearable tangible interfaces, specifically wrist-worn devices, are the most common approach used by researchers to collect workers’ behavioral and physiological data, which is used to develop machine-learning models to assess different levels of stress. However, less than 20% of the retrieved studies developed practical interventions to improve employee’s mental well-being, which shows a clear room for improvement in the development of tangible interfaces aimed at reducing work-related stress.

There also exists a trade-off between unobtrusiveness and privacy, as the data collected can be misused by companies and further induce stress in workers who are required to use such systems in real work environments, which does not positively contribute to the workplace, given that the social environment is as important to mental well-being as material safety, especially in the current climate. Another pressing issue has to do with transparency when reporting research findings, since only by understanding what has been studied is it then possible to design new and better interactive interfaces to improve employees’ mental states and create healthier workplaces.

## Figures and Tables

**Figure 1 ijerph-19-16197-f001:**
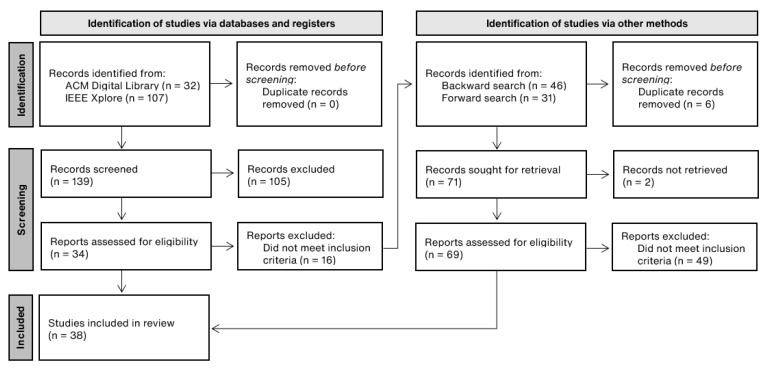
Descriptive flowchart of the systematic literature search process.

**Figure 2 ijerph-19-16197-f002:**
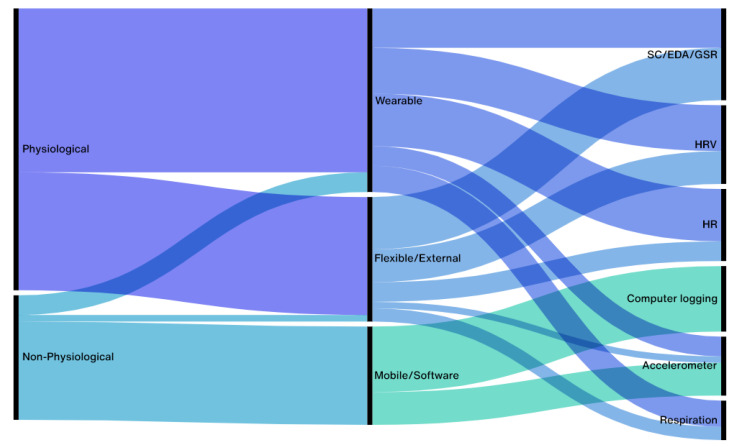
Most common stress-related measurement methods employed by the selected studies (i.e., used at least five times) by sensing modality and their relation to sensors used. SC = Skin Conductance, EDA = Electrodermal Activity, GSR = Galvanic Skin Response, HRV = Heart Rate Variability, HR = Heart Rate.

**Figure 3 ijerph-19-16197-f003:**
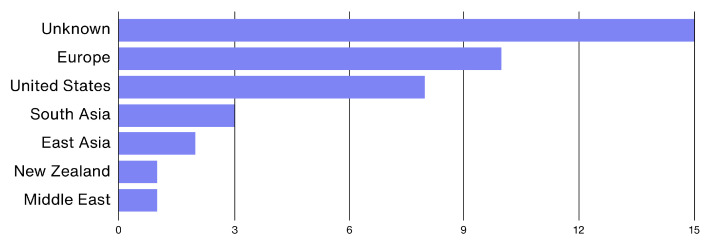
Geographic distribution of the samples used in the included studies where applicable.

**Table 1 ijerph-19-16197-t001:** Concept Matrix with the different approaches used by the selected studies to help assess and mitigate work-related stress. Monit. stands for Monitoring and Interv. for Intervention. Tangible and mobile interfaces were intentionally discriminated by the authors.

REFS.	CONCEPTS						
	**Tangible**		**Affective**		**Mobile**		**Embodied**
	**Monit.**	**Interv**.	**Monit**.	**Interv**.	**Monit**.	**Interv**.	
[26,27]							×
[28,29,30,31,32]			×				
[33]	×						
[34]		×		×			
[35,36]			×		×		
[37]			×				×
[38,39,40,41,42,43,44,45,46,47,48,49,50,51,52]	×		×				
[53,54]	×		×	×			
[55]	×	×	×	×			
[56]	×	×	×	×		×	
[57]	×		×	×		×	
[58]	×		×				
[59,60,61,62]	×		×		×		
[63]			×	×	×	×	×

**Table 2 ijerph-19-16197-t002:** Summary of the studies that contained stress-mitigating interventions and their key findings (when provided).

Ref.	Duration	Setting	Description	Key Findings
Abdul Manaf et al., 2021 [56]	Long-term	Controlled	Comparison of an immersive Virtual Reality (VR) mobile game and a non-immersive desktop version for stress reduction	No significant differences between immersive and non-immersive experiences, although both were effective at mitigating stress. Using natural scenery can helps users relax.
Bostock et al., 2019 [57]	Long-term	In the wild	Mindfulness meditation app	Short-guided mindfulness sessions improve daily affective states and continuous use increases long-term well-being.
Paredes et al., 2014 [63]	Long-term	In the wild	Mobile recommender system that uses Machine Learning (ML) to deliver personalized web-based interventions	Experience sampling (ESM) increased users’ stress awareness and created an additional source of stress. Micro-interventions are preferred, especially somatic ones that can be achieved individually.
Sano et al., 2015 [53]	Long-term	In the wild	Desktop application to deliver suggestions and recommendations on how to relieve stress	Somatic interventions were considered the most accepted and effective, whereas positive psychology and meta-cognitive ones were rated the worst
Sano et al., 2017 [54]	Long-term	In the wild	Computer software that delivers micro-stress interventions during work hours	Stress interventions have a higher acceptance rate at the beginning of the day, or at around 1 p.m. (i.e, before users become immersed in a task). Participants preferred individual and social somatic interventions to reduce stress.
Ladakis et al., 2021 [55]	Short-term	Controlled	Gamified VR approach to help mitigate stress levels in work environments	n/a
Thoondee and Oikonomou, 2017 [34]	n/a	Controlled	Virtual Reality application	Virtual environments foster relaxing states and can help improve users’ focus. The implementation of VR apps is considered acceptable in work environments.

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
