# Peer review of "From Monitoring to Assisting: A Systematic Review towards Healthier Workplaces"

_ijerph, 2022, doi:10.3390/ijerph192316197_

Round 1

Reviewer 1 Report

The paper presents a systematic review of affective computing approaches to workplace stress monitoring and management, from a technology (HCI) oriented standpoint. It attempts to discover trends and provide insight into the gaps that still need to be addressed. The paper reads reasonably well and could be good resource to researchers attempting to delve into this field, but some points, which are listed below, need to be addressed.

Stating "...the authors set out to review the last 10 years of existing literature focused on the development of technologies to monitor, evaluate and positively influence workers’ physical health and affective states." but then narrowing this to "To be included in this review, records needed to be related to the development or use of embodied, affective or tangible interfaces for predicting or delivering interventions that focus on mitigating stress-related problems in work environments." is conflicting.

In some parts, the paper is specific and gives informed hints, while in others it paints a very broad picture, not going into details which could be beneficial for both researchers and practitioners. Machine learning approaches are mentioned on multiple occasions, but no insight is given as to which approaches or data types are most effective and most commonly utilised for these. Benefits and drawbacks of utilising different data modalities should also be given some attention, especially in contrast to questionnaire-based data acquisition and light of intrusive sensors influencing the outcomes.

The authors emphasize data protection and privacy, but do not provide any detail as to how it is currently achieved or what would be good approaches or steps to take in this direction, especially if this data is to be shared among researchers.

Could you provide some examples of tangible artifacts that could be used, both for monitoring and interventional purposes? As you state that most reviewed studies do use wearable sensors (tangible interfaces), and you also state that tangible artifacts are overlooked, I suppose this is not what is meant under this term?

Reviewer 2 Report

The authors should improve the Methodology section:

- They should explain why their interest is just on Computer Science and Human-Computer Interaction (HCI) journals and conferences.

- and why haven´t they considered the PRISMA statement for reporting systematic reviews for their study?

Figure 2 should be improved, or include a better explanation of it on the main text.

To do so, the following scientific repositories and databases were searched for re- 163 search articles and journal papers published during the last 10 years: ACM Digital Library 164 (The ACM Guide to Computing Literature) and IEEE Xplore. Both portals cover most 165 of Computer Science and Human-Computer Interaction (HCI) journals and conference 166 proceedings. Even though looking at psychology journals would introduce interesting 167 research, this work focuses on HCI literature. 168

The search was done by combining the following search terms: Keyword:(stress OR 169 anxiety OR mental health OR wellbeing OR well-being) AND Keyword:(work* NOT work- 170 shop NOT workload OR office OR business OR job OR occupational OR corporat*) AND 171 Abstract:(intervention OR manage* OR regulat* OR sensing OR measure* OR recognition 172 OR detection) AND Abstract:(tangible OR affective OR embodied OR interact*). To ensure 173 the accuracy of the search process, different spellings were intentionally used, as well as 174 the asterisk (*) in the terms, which indicate a wildcard.

Round 2

Reviewer 2 Report

Thanks to the authors for considering the  review comments on their previous manuscript.

Especially the Methodology section, in its new version it is much more clarifying.

In my opinion, this version of the article is already acceptable.